# CFTR Modulators in People with Cystic Fibrosis: Real-World Evidence in France

**DOI:** 10.3390/cells11111769

**Published:** 2022-05-28

**Authors:** Lucile Regard, Clémence Martin, Espérie Burnet, Jennifer Da Silva, Pierre-Régis Burgel

**Affiliations:** 1French Cystic Fibrosis National Reference Center, Department of Respiratory Medicine, Hôpital Cochin, Assistance Publique-Hôpitaux de Paris, 75014 Paris, France; lucile.regard@aphp.fr (L.R.); clemence.martin@aphp.fr (C.M.); esperie.burnet@aphp.fr (E.B.); jennifer.dasilva@aphp.fr (J.D.S.); 2Institut Cochin, Université de Paris Cité, INSERM U1016, 75014 Paris, France; 3ERN Lung Cystic Fibrosis Network, Frankfurt, Germany

**Keywords:** cystic fibrosis, CFTR modulators, real-world studies, ivacaftor, tezacaftor, elexacaftor

## Abstract

Cystic fibrosis (CF) is a rare genetic multisystemic disease, the manifestations of which are due to mutations in the gene encoding the CF transmembrane conductance regulator (CFTR) protein and can lead to respiratory insufficiency and premature death. CFTR modulators, which were developed in the past decade, partially restore CFTR protein function. Their clinical efficacy has been demonstrated in phase 3 clinical trials, particularly in terms of lung function and pulmonary exacerbations, nutritional status, and quality of life in people with gating mutations (ivacaftor), homozygous for the F508del mutation (lumacaftor/ivacaftor and tezacaftor/ivacaftor), and in those with at least one F508del mutation (elexacaftor/tezacaftor/ivacaftor). However, many questions remain regarding their long-term safety and effectiveness, particularly in patients with advanced lung disease, liver disease, renal insufficiency, or problematic bacterial colonization. The impact of CFTR modulators on other important outcomes such as concurrent treatments, lung transplantation, chest imaging, or pregnancies also warrants further investigation. The French CF Reference Network includes 47 CF centers that contribute patient data to the comprehensive French CF Registry and have conducted nationwide real-world studies on CFTR modulators. This review seeks to summarize the results of these real-world studies and examine their findings against those of randomized control trials.

## 1. Introduction

Cystic fibrosis (CF) is the most common genetic disease in Caucasian populations and currently affects more than 100,000 individuals worldwide [1,2]. It was first described in the 1930s in autopsy studies showing cysts and fibrosis of the pancreas in malnourished infants [3] and was rapidly recognized as a genetic autosomal disorder [4]. The disease is caused by the presence of mutations in the gene encoding for the CF transmembrane conductance regulator (CFTR) protein, a chloride and bicarbonate ion channel expressed at the surface of epithelial cells [5]. This ion transport defect results in a multisystem disease affecting the lungs, the pancreas, the intestinal tract, the liver, and reproductive organs. More than 2000 mutations in the *CFTR* gene have been described and are divided into six functional classes (Figure 1): class I, II, and III disease-causing mutations are associated with little to no CFTR function (usually associated with a more severe phenotype), while class IV, V and VI mutations maintain residual CFTR function (usually associated with milder phenotype) [6]. F508del, a class II mutation, is the most common; approximately 80% of people with CF (pwCF) carry one F508del mutation, and 40–50% of patients are homozygous for this mutation [6,7].

In the 1950s, the life expectancy of newborns with CF barely reached a year, with meconium ileus and malnutrition being the main causes of death [4]. Although CF remains a fatal condition, with lung disease being the major cause of morbidity and mortality [8], significant improvement in survival has been achieved in the past decades, with a current median life expectancy over 50 years of age [6]. This remarkable increase in life expectancy and quality of life can be attributed to multidisciplinary care in CF centers, neonatal screening, nutritional support, antibiotic therapy (e.g., *Pseudomonas aeruginosa* eradication, early treatment of acute pulmonary exacerbations), intensive physiotherapy for mucus clearance, mucoactive drugs, and treatment of CF-related complications [9]. As a result, the demographic characteristics of the CF population have dramatically changed, and CF is no longer considered a strictly pediatric disease since adults (18 years and older) represent 50% to 60% of patients in countries with well-established CF care [1]. Between 2010 and 2025, the overall number of pwCF is expected to have increased by approximately 50%, with a 20% increase in children with CF and 75% in adults [10,11]. These forecasts were made at a time when targeted CFTR modulator therapy was not available.

CFTR modulators are small molecules that bind to defective CFTR proteins and partially restore their function [12]. Four CFTR modulators have been approved for use in pwCF (Table 1), though access and eligibility criteria vary across countries. Ivacaftor (IVA), a CFTR potentiator and the first to have been developed, is now approved for patients aged 4 months and older with gating mutations (class III) in Europe and has been used for over a decade [13]. The promising results of IVA encouraged the development of CFTR correctors lumacaftor (LUM) and tezacaftor (TEZ), used in combination with IVA to target F508del, the most common *CFTR* mutation. In France, both combinations are approved in patients homozygous for the F508del mutation [14,15], and TEZ/IVA is also approved for patients carrying one copy of the F508del mutation and selected residual function mutation [16]. Finally, elexacaftor (ELX), a next-generation corrector combined with tezacaftor and ivacaftor, was approved in 2020 for patients homozygous for the F508del mutation or presenting a the F508del mutation associated with a minimal function, gating, or residual function mutation [17,18,19]. Approximately 82% of pwCF are known to carry the F508del mutation and are therefore eligible to receive elexacaftor/tezacaftor/ivacaftor (ELX/TEZ/IVA) (Figure 2).

CFTR modulators have dramatically changed clinical care, introducing a fundamental shift in perspective for pwCF and their caregivers. However, the promising safety and efficacy results were obtained in clinical trials with a limited number of participants and strict inclusion criteria: moderate to severe respiratory impairment (percent predicted forced expiratory volume in 1 sec -ppFEV_1_-between 40–90), normal renal and hepatic function, or no *B. cepacia* or *M. abscessus* colonizations. These selection criteria limit the generalizability of the results of these clinical trials, particularly for patients with advanced CF lung disease (ppFEV_1_ < 40), those with multiple comorbidities such as renal or hepatic failure, and/or problematic bacterial colonization (e.g., *B. cepacia*, *M. abscessus*). In addition, due to their short duration, the randomized controlled trials (RCTs) evaluating safety and efficacy could not include long-term treatment outcomes or the impact of CFTR modulators on variables such as mortality, lung transplantation, changes in concurrent therapies, lung clearance index, or chest imaging in their analyses.

These questions are generally addressed in real-world studies. In France, the great majority of pwCF attend one of the 47 CF care centers, which are distributed throughout the country and organized in a national reference network (Figure 3). The CF centers and the transplantation centers contribute to the French CF Registry, which captures data on more than 95% of the CF population. This registry was adapted to conduct real-world studies in adults and adolescents with CF treated with approved CFTR modulators.

The aim of the present manuscript is to review, for each approved modulator: (1) the results of the main phase 3 randomized controlled trials that evaluated the safety and efficacy of CFTR modulators in adolescents (12–17 years) and adults (18 years and older) with CF (summarized in Table 2); (2) the findings of real-world studies conducted in France through the French CF reference center network in adults and adolescents with CF (summarized in Table 3).

The association TEZ/IVA was approved only recently in France, and in a very limited number of patients, most of whom are now eligible for ELX/TEZ/IVA and are therefore not included in this review.

## 2. Ivacaftor

### 2.1. Main Randomized Controlled Trials

Ivacaftor was the first CFTR modulator to be approved in France (commercial name Kalydeco^®^) for the treatment of pwCF who carry at least one gating mutation (Table 1). IVA is a potentiator that binds to the CFTR protein in the plasma membrane and increases the CFTR channel’s opening frequency and ion conductance [24,25]. The first phase 3 trial enrolled 161 patients aged 12 and older with at least 1 G551D mutation and a ppFEV_1_ between 40–90. After 48 weeks of treatment with IVA, respiratory function, nutritional status, and quality of life improved significantly, while sweat chloride concentrations and pulmonary exacerbation frequency decreased [13]. A subsequent study in children aged 6–11 years with at least 1 G551D mutation showed a significant increase in ppFEV_1_ and weight gain and a reduction in sweat chloride concentrations [26]. However, there was no significant improvement in the Cystic Fibrosis Questionnaire-Revised (CFQ-R) score or in the frequency of pulmonary exacerbations. Three years later, De Boeck et al. assessed the efficacy of IVA in patients six years and older with a non-G551D gating mutation [21]. They reported a significant increase in ppFEV_1_, body mass index (BMI), and CFQ-R scores in the IVA group compared to placebo, along with a decrease in sweat chloride concentrations, suggesting that IVA could also be effective with non-G551D gating mutations. In the KONDUCT trial, Moss et al. explored its safety and efficacy in pwCF aged 6 years and older with at least one copy of the R117H-*CFTR* mutation (about 3% of the CF patient population), which is associated with both gating and conductance function [22]. No significant difference was reported in ppFEV_1_ or BMI after 24 weeks of treatment, though significant improvements were found in CFQ-R score and sweat chloride concentrations in the IVA group. Interestingly, a subgroup analysis showed a clear increase in ppFEV_1_ in patients over 18 years of age receiving IVA. These results may have been attributable to the fact that children with the R117H-*CFTR* mutation generally have a milder phenotype and delayed pulmonary involvement [22]. In children under the age of 6, the KIWI and the ARRIVAL trials confirmed that IVA was safe, well-tolerated, and resulted in significant improvement in sweat chloride concentrations and nutritional status, but not in lung function [27,28]. In both studies, the authors showed that IVA was associated with improvement in fecal elastase-1, suggesting that IVA, if started early, may preserve exocrine pancreatic function.

**Table 3 cells-11-01769-t003:** French real-world studies on CFTR modulators.

Modulator	Author Year	Outcomes	Follow-Up Duration	*n*	Genotype	Age (Years) Mean [Range] *% <18-Year-Old*	ppFEV_1_	Main Findings
**Ivacaftor**	Hubert 2021 [29]	Effectiveness and healthcare resource utilization of IVA in pwCF *Prospective*	24 mo	129	At least one gating or R117H mutation	19.1 [2–64] *58.9%*	75.2 (±24.9)	- ppFEV1 increased by a least-squares mean of 8.49 percentage. - points in the first 6 months and sustained through 36 months. - Growth metrics increased during the first 12 months post-IVA and remained stable. - Decrease in the rate of PEx during the 12 months post-IVA. - Decrease in healthcare resource utilization. - No new safety concerns identified; discontinuation: 5.6%.
	Chassagnon 2016 [30]	Short-and long-term HRCT changes in adult pwCF treated with IVA *Retrospective*	8–33.1 mo	22	At least one gating mutation	36.0 [NA] *NA*	31.5–77.0	- CT scan = valuable method for monitoring CF patients treated with IVA. - Decreased mucus plugging and peribronchial thickening during the first year, stable over long-term follow-up. - Bronchiectasis score slightly increased, possibly due to improved visualization, after mucus plugging clearance. - Moderate correlation between interscan changes in FEV1 and CT scores.
	Sermet 2016 [31]	Impact of IVA on bone mineralization *Retrospective*	1–3 yr	7	At least one G551D mutation	37 (median) [26–52] *0%*	48.0 (±9)	- Improved bone mineral density in pwCF carrying the G551D mutation.
	Hubert 2018 [32]	Clinical response to IVA in pwCF aged 6 or older *Retrospective*	12–24 mo	57	At least one G551D mutation	17.6 [6–52] *53%*	72.3 (±26.4)	- Improvement in ppFEV1 from baseline to Year 1 (+8.4%; *p* < 0.001) and Year 2 (7.2%; *p* = 0.006). - Statistically significant increase in BMI, fewer Pseudomonas aeruginosa and Staphylococcus aureus positive cultures, decrease in IV antibiotics and maintenance treatment. - No significant adverse events reported; discontinuation: 3.5%.
**Lumacaftor** **+** **Ivacaftor**	Masson 2019 [33]	Factors involved in the individual’s response to LUM/IVA *Prospective*	6 mo	41	Homozygous for F508del	15.7 [NA] NA	68.2 (±3.6)	- Increased ppFEV1: +5%. - Increased BMI: +3.7%. - Sweat chloride: −20 mmol/L. - In vivo biomarkers of CFTR activity (sweat chloride, nasal potential difference, intestinal short-circuit current measurements) not correlated with the improvements in clinical status. - LUM and IVA blood levels not predictive of the clinical response.
**Lumacaftor** **+** **Ivacaftor**	Misgault 2020 [34]	Impact of LUM/IVA on glucose tolerance abnormalities *Prospective*	12 mo	40	Homozygous for F508del	24 [12–61] *45%*	61 (48–78)	- Proportion of patients with glucose intolerance decreased from 78% to 40%. - Proportion of patients with diabetes decreased from 22% to 10%. - Improved glucose tolerance in 57.5% with a significant decrease in both 1-h and 2-h OGTT glycemia.
	Bui 2021 [35]	Clinical, radiological and metabolic response to LUM/IVA and factors associated with response to treatment in CF adolescents *Prospective*	24 mo	40	Homozygous for F508del	13.9 [NA] *100%*	83.3 (±18.3)	- Improvement in ppFEV1 (+5.8%). - Significant improvement in BMI Z-score and sweat chloride concentrations. - No significant change in exacerbation rates, antibiotic use, or CT scan scores. - Lower age associated with better response and greater ppFEV1 change. - Discontinuation: 0%.
	Campredon 2021 [36]	Lung structural changes in pwCF treated with LUM/IVA and morphological phenotypes associated with response to treatment *Prospective*	12 mo	283	Homozygous for F508del	23.4 [NA] *39%*	65.9 (±19.6)	- Significant decrease in the Bhalla score (−1.40 ± 1.53 points). - Significant decrease in mucus plugging, bronchial wall thickening, and parenchymal consolidations. - Identification of a subgroup of patients with milder structural lung abnormalities at baseline, which predicted higher rate of ppFEV1 response to LUM/IVA. - No significant correlation between morphological improvement and ppFEV1 improvement.
	Hubert 2017 [37]	Short-term AEs and effectiveness of LUM/IVA in adults with severe lung disease *Retrospective*	3 mo	53	Homozygous for Phe508del	31.1 [20–48] *0%*	31.9 (±5.4)	- Respiratory AEs reported in 51% of patients. - Discontinuation: 30%. - ppFEV1 + 2.06 after 1 month and +3.19 after 3 months. - BMI unchanged.
	Burgel 2020 [38]	Safety and effectiveness of LUM/IVA in adolescents and adults. *Prospective*	12 mo	845	Homozygous for Phe508del	22.0 (median) [16–30] *34.6%*	65.0 (47–80)	- Increased ppFEV1: +3.67% - Increased BMI: +0.73 kg/m^2^ - Decrease in IV antibiotic courses: −35%. - Discontinuation: 18.2%, owing to adverse events. - Factors associated with discontinuation: adult age group, ppFEV1 < 40, and numbers of IV antibiotic courses in the year prior to LUM/IVA initiation. - After treatment discontinuation: decrease in ppFEV1, no BMI improvement, no decrease in the number of IV antibiotic courses.
**Lumacaftor** **+** **Ivacaftor**	Olivereau 2020 [39]	Adherence and factors associated with adherence in patients treated with LUM/IVA *Retrospective*	12 mo	96	Homozygous for Phe508delF508del	22.0 [NA] *55%*	77.0 (±25)	- Adherence defined as ≥80% days covered, using pharmacy refill data. - Adherent patients: 89% and 83% at 6 and 12 months, respectively. - Probability of being adherent increased with age andppFEV1. - Higher adherence than other CF therapies.
	Tétard 2020 [40]	Intestinal inflammation (fecal calprotectin concentrations) in CF adolescents treated with LUM/IVA *Retrospective*	336 days	15	Homozygous for Phe508delF508del	12 [12–16] *100%*	89.0 (71–99.5)	- Significant decrease in fecal calprotectin concentrations from 713 mg/g to102 mg/g. - Significant decrease in intestinal inflammation. - Decrease of intestinal inflammation not correlated with respiratory function changes.
	Burgel 2020 [41]	Clinical response to LUM/IVA according to baseline lung function *Prospective*	12 mo	827	Homozygous for F508del	**ppFEV_1_ < 40** 30 (median) [NA] *12.4%* **ppFEV_1_ [40–90]** 21 (median) [NA] *40.6%* **ppFEV_1_ ≥ 90** 20 (median) [NA] *28.9%*	33.7 (30.9–36.9) 66.2 (53.6–76.9) 96.4 (93.0–101.7)	- Significant increase in ppFEV1 for patients with ppFEV1 [40–90] (+2.9%,) and those with ppFEV1 < 40 (+0.5%), but not in those with ppFEV1 ≥ 90. - Number of days of IV antibiotics reduced in all subgroups. - Comparable increase in BMI for all subgroups. - Discontinuation rate higher in ppFEV1 < 40 patients (28.9%) than in those with ppFEV1 [40–90](16.4%) or ppFEV1 ≥ 90 (17.5%).
	Arnaud 2021 [42]	CT changes in pwCF treated with LUM/IVA *Retrospective*	15.4 mo (7–54 mo)	33	Homozygous for F508del	26.0 [12–58] *62%*	74.8 (±27.0)	- Significant decrease in Brody score and mucous plugging subscore. - Peribronchial wall thickening significantly improved in adults. - Improvements in CT scores significantly correlated with ppFEV1.
	Reix 2021 [43]	LCI evolution in pwCF treated with LUM/IVA and its clinical value compared to ppFEV_1_ *Retrospective*	6–12 mo	63	Homozygous for F508del	16 (median) [12–20] *96.8%*	72.8 (59.6–80.7)	- At both M6 and M12, no statistically significant LCI increases (worsening) of 0.13 units and 0.6 units. - Discordant results between LCI and ppFEV1 in one-third of the patients.
** *Elexacaftor* ** ** *+* ** ** *Tezacaftor* ** ** *+* ** ** *Ivacaftor* **	Burgel 2021 [44]	Safety and effectiveness of ELX/TEZ/IVA in pwCF with advanced respiratory disease *Prospective*	3 mo	245	At least one F508del mutation	31 (median) [NA] *6.9%*	29 (24–34)	- Rapid improvement in ppFEV1 (+15.1%) and weight gain (+14.2 kg). - Significant reduction in the need for long-term oxygen, non-invasive ventilation, and/or enteral tube feeding (respectively, 50%, 30%, and 50%). - Indication for lung transplantation suspended for most patients on the transplant waiting list or undergoing transplantation evaluation. - Compared with the previous 2 years, a 2-fold decrease in the number of lung transplantations was observed in 2020, with no concurrent increase in deaths without transplantation. - No discontinuation; AEs generally mild.
	Martin 2021 [45]	Perceived changes in respiratory symptoms, systemic manifestations, treatment burden, and impact on quality of life in pwCF treated with ELX/TEZ/IVA *Prospective*	6 days–7.3 mo	101	At least one F508del mutation	35 (median) [NA] *3.0%*	NA	- Significant improvement in respiratory symptoms, sleep quality, and physical self-esteem. - Reduction in treatment burden (chest physiotherapy, IV antibiotic courses, hospitalizations, diabetes control, other treatments, lung transplant discussions). - Positive physical and psychological effects translated into improved quality of life, new life goals, and overwhelmingly positive impact on general well-being.
	Martin 2022, [46]	Impact of ELX/TEZ/IVA on lung transplant candidates: lung transplantation status, clinical findings, healthcare utilization, and concurrent treatments *Prospective*	12 mo	65	At least one F508del mutation	32 (median) [14–65] *4.6%*	25 (21–30)	- A total of 17 patients listed for transplantation, and 48 considered for listing within 3 months at baseline. - After 1 year, 2 patients transplanted, 2 listed for transplantation, and 61 no longer met transplantation criteria. - AEs generally mild, no discontinuation. - Rapid and sustained increase in ppFEV1 (+13.4%) and BMI (2.6 kg/m^2^). - Significant reduction in IV antibiotic courses, hospitalizations, and need for oxygen therapy and non-invasive ventilation.

mo: months; BMI: body mass index; PEx: pulmonary exacerbation; CT computed tomography; ppFEV_1_: percent predicted forced expiratory volume in 1 s; IV: intravenous. AEs: adverse events; OGTT: oral glucose tolerance test; LCI: lung clearance index. NA: not available. For ppFEV_1_, data are expressed as mean (±SD), mean (SEM) or median (IQR or range) depending on study data.

Finally, the DISCOVER trial explored the effects of a 16-week course of IVA in patients homozygous for the F508del mutation and found no change in ppFEV_1_, BMI, or CFQ-R score; a small reduction in sweat chloride concentrations was reported but was not sustained throughout the 96-week study period [47].

These RTCs had a study duration of 48 weeks or less and excluded patients with advanced lung disease (i.e., ppFEV_1_ < 40), abnormal liver function tests, or renal failure. Serious adverse events (AEs) were reported in 10.5 to 24% of patients receiving IVA, the majority of which were consistent with CF disease manifestations rather than with drug-related AEs: pulmonary exacerbations, cough, upper respiratory infections, nasal congestion, and chest tightness or diarrhea. Furthermore, the incidence of AEs was similar in the IVA and placebo groups in most trials [13,21,22,25,47], and the proportion of patients who discontinued treatment was, in fact, lower in the IVA groups, ranging from 0 to 7.7%. Of note, a higher incidence of elevated liver function test (LFT) was reported in the pediatric studies than in adult ones [48].

### 2.2. Ivacaftor in the French Real-World Studies

#### 2.2.1. Safety and Effectiveness

In 2018, Hubert et al. published the first retrospective study using the French CF registry to assess the long-term safety and effectiveness of IVA in 57 children and adult pwCF carrying at least one copy of the G551D mutation [32]. They confirmed the increase in ppFEV_1_ and weight that were reported in RCTs. Moreover, IVA was associated with a decrease in *P. aeruginosa* and *Staphylococcus aureus* colonization and with fewer IV antibiotic courses and maintenance treatment prescriptions (including azithromycin, dornase alpha, and nutritional supplements). These changes were sustained after 2 years, and no safety alert was reported. The mean absolute increase in ppFEV_1_ in this study was lower than that reported in the RCTs [4,13]; it was comparable to that found in real-world studies on IVA conducted in other countries [49,50]. The ppFEV_1_ increase in patients with advanced lung disease (i.e., ppFEV_1_ < 40) was lower in comparison to the total registry population (5% vs. 8.3%) but was comparable to the findings of studies conducted in Germany and the United States [51,52].

These results were confirmed in the BRIO study, a prospective observational study conducted in 35 centers in the French CF network, which included 129 pwCF with gating mutations (40.1% children <12 years) and was designed to assess the effectiveness of IVA in pwCF in terms of clinical outcomes and healthcare resource utilization within 2 years after IVA initiation [29]. The main results included substantial improvements in ppFEV_1_ (least-squares mean of 8.49 percentage points), growth metrics, and nutritional status within the first 6–12 months and up to 36 months after IVA initiation [29]. The rate of pulmonary exacerbations (PEx) decreased during the 12 months post-ivacaftor in comparison to the 12 months prior, with an estimated risk ratio (RR) of 0.57 (95% Confidence Interval (95%CI) 0.43–0.75) for PEx events and 0.25 (0.13–0.48) for PEx requiring hospitalization [29]. Hubert et al. also observed a decrease in the proportion of pwCF who had a positive sputum culture over 12 months and lower methicillin-sensitive *S. aureus* and *P. aeruginosa* colonization rates [29]. Little to no difference was observed between the pre-and post-ivacaftor periods in terms of classic CF comorbidities (diabetes, exocrine pancreatic insufficiency) or in medication requirements. IVA was generally well-tolerated, and no deaths occurred [29].

The results obtained in the French CF population are comparable to those recently published by Volkova et al., who used data from the United States and United Kingdom CF registries to compare patients treated with IVA to non-eligible patients [53]. The authors showed that the IVA group had better lung function and nutritional status and lower PEx rates after 5 years [53]. In a recent report on 35 patients treated with IVA for 5 years, Mitchell et al. reported comparable findings with an initial increase in ppFEV_1_ and a reduction in hospitalizations and intravenous antibiotic use. They also showed that lung function decline was unaltered, however, raising questions about the long-term history of CF with highly effective CFTR modulators [54].

#### 2.2.2. Morphological Changes

Computed tomography (CT) changes in adult pwCF treated with CFTR modulators were not evaluated in RCTs. Chassagnon et al., using the CF-CT score on the whole lung following volumetric chest CT examination, reported both short- (3–18 months) and long-term (>18 months) improvements with IVA [30]. These improvements were attributable to a significant decrease in peribronchial thickening and mucus plugging, both of which are commonly seen during exacerbations. A significant increase in the bronchiectasis score was also observed after 18 months, probably due to improved visualization after the decrease in bronchial thickening and mucus plugging [30]. Most of the CT changes occurred during the first year of treatment and remained stable over the long term, which was also reported with lung function [30,53].

#### 2.2.3. Bone Disease

Sermet-Gaudelus et al. showed that IVA improved CFTR-related bone disease in 7 adult patients carrying the G551D mutation whose lumbar spine z-score increased by a mean of 0.9 (0.1–2.5) after 1.7 years of treatment on average (*p* = 0.04) [31]. These results were not confirmed in Hubert et al.’s retrospective study, however [32].

## 3. Lumacaftor/Ivacaftor

### 3.1. Main Randomized Controlled Trials

Lumacaftor (LUM) was the first CFTR corrector acting on the F508del-*CFTR* mutation to be approved. Its mechanism of action is not fully understood, but several studies suggest that it could repair the aberrant assembly of the full-length protein and improve its processing, trafficking, and stability [55]. It was approved in France for patients aged 2 years and older homozygous for the F508del mutation in combination with ivacaftor (LUM/IVA, commercial name, Orkambi^®^) in 2019 (Table 1).

The first RCTs to assess the safety and efficacy of LUM/IVA in patients homozygous for the F508del mutation were conducted over 28 days in patients 18 years and older with a ppFEV_1_ between 40–90 (phase 2) [56] and over 24 weeks in patients older than 12 years with a ppFEV_1_ ≥ 40 (Phase 3) [14]. Both studies showed modest but significant positive effects on lung function, BMI, pulmonary exacerbations, and hospitalizations [14,56] (Table 2). Participants in these 2 studies were then eligible to enroll in an open-label 48 week-trial which showed that clinical outcomes were sustained over time, confirming the long-term benefits of the LUM/IVA combination in F508del homozygous patients [57]. RCTs evaluating LUM/IVA in patients with only one F508del mutation showed no clinical benefit [56,58].

Subsequent phase 3 studies in children under the age of 11 years and homozygous for the F508del mutation consistently confirmed a significant improvement in lung function. Their findings also showed that LUM/IVA was associated with a decrease in LCI2.5 and sweat chloride concentrations and with improvements in nutritional status or in growth metrics and health-related quality of life. It was generally well-tolerated [59,60].

Overall, the incidence of AEs was comparable in the LUM/IVA and placebo groups, with a rate of serious AEs ranging from 15.4 to 45%. The most common AEs were pulmonary exacerbations, cough, headache, dyspnea, chest tightness, hemoptysis, and increased sputum production. Discontinuation rates were low (5.6–8.1%) in all three trials [14,56,58]. However, in the open-label extension study, dyspnea and chest tightness were reported more frequently, and the discontinuation rate was higher in the LUM/IVA group compared to the placebo group [57].

Another combination of a CFTR corrector and potentiator, Tezacaftor/Ivacaftor (TEZ/IVA), was approved in Europe in pwCF aged 6 years and older, homozygous or heterozygous for the F508del mutation. TEZ/IVA was evaluated in the 24-week phase 3 EVOLVE trial, which enrolled CF patients 12 years and older homozygous for the F508del mutation [15]. The authors reported a +4.0% increase in ppFEV_1_ as well as higher CFQ-R scores and a decrease in the rate of pulmonary exacerbations (−35%) and in sweat chloride concentrations. The improvement in lung function was comparable to that observed in the phase 3 TRAFFIC/TRANSPORT studies on LUM/IVA, but TEZ/IVA had a better tolerance profile with a lower incidence of dyspnea, chest tightness, and other respiratory symptoms. A recent report showed that TEZ/IVA was generally safe, well-tolerated, and effective for up to 120 weeks in patients aged 12 years and older homozygous for the F508del mutation or heterozygous with a residual function mutation [61].

### 3.2. Lumacaftor/Ivacaftor Combination in the French Real-World Studies

#### 3.2.1. Safety and Effectiveness

The first real-world study on LUM/IVA conducted in the French CF Reference network was published in 2017 by Hubert et al. and evaluated the short-term safety and effectiveness of LUM/IVA in 53 adults with CF and severe lung disease [37]. The authors reported that respiratory AEs occurred in 51% of included patients and resulted in treatment discontinuation in 24%, which was markedly higher than the 5% or fewer rates of discontinuation found in phase 3 clinical trials [14]. On average, patients who were able to continue LUM/IVA had a 3.19 point-higher ppFEV_1_ at 3 months, even when their ppFEV_1_ was ≤30 at baseline, which was comparable to what was observed in patients recruited in phase 3 clinical studies [14].

Three years later, Burgel et al. (2020) confirmed and extended these results in 845 adolescents and adults homozygous for the F508del mutation [38]. They showed that patients who were able to continue LUM/IVA over one year had improved lung function (ppFEV_1_ +3.67) and BMI (+0.73 kg/m^2^) and fewer pulmonary exacerbations requiring antibiotic courses (−35%). These results were comparable to what was observed in the RCTs [38]. However, the proportion of patients who discontinued LUM/IVA was more than three times higher compared to phase 3 studies (18.2% vs. 5%); and reached 30% in patients with a ppFEV_1_ < 40 [38]. Treatment discontinuation was mostly related to the respiratory adverse events that were observed with lumacaftor (though not with tezacaftor) [62]. Factors associated with discontinuation were age group ≥ 18 years old, ppFEV_1_ < 40, and more IV antibiotic courses during the year prior to LUM/IVA initiation. These findings were consistent with a phase 3b open-label prospective study evaluating LUM/IVA in patients with advanced lung disease (ppFEV_1_ < 40), which reported more frequent respiratory AEs and recommended treatment initiation at a lower dose [63].

Examining the effects of LUM/IVA at various levels of baseline lung function in 827 pwCF, one-third of whom had a ppFEV_1_ that was either too low (<40) or too high (≥90) to meet eligibility criteria for phase 3 clinical trials, Burgel et al. (2021) reported a 1.5 to 2-fold greater increase in lung function among those with baseline ppFEV_1_ [40–90] compared to the other groups [41]. An increase in BMI was found in all patients, with a comparable magnitude of improvement across all subgroups [41]. The number of days of IV antibiotics was lower in all subgroups, but exacerbation rates remained stable in patients with severe respiratory impairment. These findings highlight the clinical benefits that can be achieved at different degrees of clinical severity, although the impact of LUM-IVA appears to vary depending on baseline lung function.

#### 3.2.2. Factors Associated with Response to Treatment

In addition to these large real-world cohort studies, smaller studies were also conducted within the French CF Center Reference Network to assess outcomes that had not been evaluated in phase 3 RCTs.

Masson et al. used CFTR biomarkers in the patient’s sweat gland, nasal and rectal mucosa, as well as serum drug concentration and the CFTR genetic context to evaluate individual response to LUM/IVA in a prospective study that recruited 41 patients aged 12 and older [33]. They did not find any correlation between clinical status improvement (ppFEV_1_, BMI z-score) and in vivo CFTR biomarkers after 6 months of treatment with LUM/IVA. Lumacaftor and ivacaftor serum levels were also not predictive of clinical response [33]. In 2021, Bui et al. prospectively investigated the clinical, radiological, and metabolic response of LUM/IVA over 24 months in 40 adolescents with CF [35]. Lung function was found to increase by +5.8 percentage points 2 years after treatment initiation, while BMI Z -score and sweat chloride concentrations improved and were sustained over 24 months. These results were particularly marked in patients with ppFEV_1_ < 80 and/or BMI z-score < 0. Age at LUM/IVA initiation was lower in good responders and was associated with a greater ppFEV_1_ change during the 2 years of treatment. There were no significant changes in exacerbation rates, antibiotic use, or CT scan scores [35].

#### 3.2.3. Morphological Changes

Chest CT scan changes were also evaluated in two imaging studies published in 2021. Arnaud et al. retrospectively reviewed the chest CT scans of 33 CF adults and adolescents before and after a mean of 15.4 months of LUM/IVA therapy, using the modified Brody score [42]. They reported a significant improvement in the total Brody score (65.5 vs. 60.3, *p* = 0.049) as well as a decrease in the mucous plugging score (12.3 vs. 8.7, *p* = 0.009). They observed a reduction in peribronchial wall thickening, which was correlated with improvements in ppFEV_1_, as were improvements in the total Brody score and mucous plugging [42]. These results were confirmed, though to a lesser degree, in a prospective multicenter study that evaluated the chest CT scans of 283 adults and adolescents with CF at baseline and 1 year after LUM/IVA initiation [36]. Increased lung function was associated with a decrease in the Bhalla score (−1.40 ± 1.53 *p* < 0.001), mucus plugging, bronchial wall thickening, and parenchymal consolidations [36]. The authors also found that ppFEV_1_ values were correlated with visual CT scores of disease severity (R = −0.51) but, contrary to Arnaud et al. [42], not with morphological improvement. A subgroup of patients had fewer structural lung abnormalities at LUM/IVA initiation and higher rates of ppFEV_1_ a year later.

#### 3.2.4. Glucose Tolerance Abnormalities

Given the lack of evidence regarding the effects of CFTR modulator therapy on glucose tolerance abnormalities (GTA), Misgault and colleagues followed 40 pwCF with GTA homozygous for the F508del (78% had impaired glucose tolerance and 22% had CF-related diabetes) before and after treatment with LUM/IVA [34]. After 1 year of treatment, 50% had normal glucose tolerance, 40% glucose intolerance, and 10% diabetes (*p* < 0.001). The 2 h oral glucose tolerance test (OGTT) serum glycemia value decreased from 171 mg/dL on average (IQR: 153–197) to 139 (117–162) mg/dL (*p* < 0.001). Overall, glucose tolerance improved in 57.5% of patients with a significant decrease in both 1 h (*p* < 0.01) and 2 h (*p* < 0.001) OGTT glycemia.

#### 3.2.5. Treatment Adherence

Adherence to LUM/IVA was evaluated at 6 and 12 months by Olivereau et al. in a retrospective study that included 96 children and adults with CF [39]. The authors examined pharmacy refill data to calculate the proportion of days covered (PDC) and defined adherence as a PDC ≥80%. Their results showed high adherence rates with a mean PDC of 96% ± 14 at 6 months and 91% ± 17 at 12 months, and a proportion of adherent patients of 89% and 83% at 6 and 12 months, respectively. This study was the first to evaluate adherence to LUM/IVA in pwCF, and the reported adherence rates were higher than those found with IVA in patients with gating mutations in a previous study [39].

#### 3.2.6. Other Outcomes

A small pilot study conducted in 2020 evaluated the effects of LUM/IVA on abdominal inflammation using fecal calprotectin concentration measurements [40]. The authors reported a substantial decrease in fecal calprotectin concentrations in 15 adolescents with CF treated for a mean of 336 days with LUM/IVA. This decrease in intestinal inflammation was not correlated with respiratory function changes, suggesting that CF-related digestive disorders may evolve independently from pulmonary disease [43]. The long-term consequences of such a decrease in intestinal inflammation have yet to be determined, particularly in terms of the risk of colorectal cancer risk in pwCF.

The effects of LUM/IVA on the lung clearance index (LCI) were evaluated in a retrospective study by Reix et al., who also examined the clinical value of using LCI to assess lung function in comparison with ppFEV_1_ at 6 and 12 months [43]. Their results showed no improvement in LCI at either time point and a mismatch between LCI and ppFEV_1_, suggesting that LCI is not an ideal outcome measure to evaluate CFTR-modulator effectiveness in adolescent and young adult pwCF with more advanced lung disease but could provide interesting indications in younger populations with milder disease [43]. These results are, however, somewhat different than recently published studies. Indeed, in a cohort of 49 patients, Shaw et al. reported an improvement in LCI 6 and 12 months after LUM/IVA initiation but not in ppFEV_1_ [64]. These differences in results between both studies may be due to marked dissimilarities in study populations. In the French cohort, patients had more severe disease and lower lung function (medians ppFEV1 at 72.8% vs. 91.3%). In their study, Graeber et al. also reported a significant improvement in LCI but not in ppFEV_1_ in 30 F508del homozygous pwCF 12 years and older before and 8–16 weeks after initiation of LUM/IV, suggesting that LCI could be more sensitive than ppFEV_1_ to detect response to CFTR modulator therapy [65]. Finally, in line with the results of the French study, Donaldson et al. reported no significant change in LCI in 25 pwCF homozygous for the F508del mutation [66].

Despite these encouraging results, corrector/potentiator combinations (i.e., LUM/IVA and TEZ/IVA) fail to completely restore CFTR protein function and are ineffective in patients heterozygous for F508del with a minimal function mutation (one that produces no protein and/or does not demonstrate in vitro response to modulators) [67]. This led to the development of next-generation correctors, which target different CFTR sites, and to triple-combination therapy associating the next-generation corrector elexacaftor (ELX) with TEZ and IVA. TEZ was preferred to LUM due to its more favorable pharmacological profile, including lower CYP3A activation [68].

## 4. Elexacaftor/Tezacaftor/Ivacaftor

### 4.1. Main Randomized Controlled Trials

Elexacaftor increases the number of mature CFTR proteins at the cell surface and was first evaluated in combination with TEZ and IVA in a 4-week phase 2 double-blind RCT that recruited CF patients 18 and older, with a ppFEV_1_ between 40 and 90, homozygous for the F508del mutation or carrying F508del and a minimal function mutation [19]. In this proof-of-concept clinical trial, ELX/TEZ/IVA was shown to increase ppFEV_1_ in both mutation groups compared to placebo as early as week 2 of treatment [19]. A decrease in sweat chloride concentrations and improvement in the respiratory domain of the CFQ-R score were also reported [19]. French health authorities have approved the use of ELX/TEZ/IVA for patients aged 12 years and older with at least one F508del mutation under the name Kaftrio^®^-Kalydeco^®^ (Table 1).

Following the promising results of this phase 2 study, Heijerman et al. conducted a 4-week phase 3 RCT in F508del homozygous patients aged 12 years and older with a ppFEV_1_ between 40 and 90 comparing ELX/TEZ/IVA with TEZ/IVA (Table 2) [17]. At week 4, in comparison to the TEZ/IVA group, patients receiving ELX/TEZ/IVA had a 10-point greater increase in ppFEV_1_. Their sweat chloride concentrations had decreased by 45.1 mmol/L, with a mean value below the diagnostic threshold for CF. The CFQ-R respiratory domain improved by 17.4 points, exceeding the 4-point improvement mark commonly used to demonstrate clinical effectiveness in CF [17]. They also had a greater increase in BMI.

A phase 3 RCT by Middleton et al. compared ELX/TEZ/IVA with placebo in 403 CF patients aged 12 and older heterozygous for F508del with a minimal function mutation [18]. By week 4, ppFEV_1_ had increased by 13.8 points, and the effect was sustained throughout the 24-week study period. The authors also reported an increase in the CFQ-R respiratory domain score of 20.2 points, a 63% reduction in the rate of pulmonary exacerbations, and a decrease of 41.8 mmol/L in sweat chloride levels. BMI also improved significantly (Table 2).

The patients who participated in both phase 3 trials were invited to enroll in a phase 3 open-label extension study on the long-term safety and efficacy of ELX/TEZ/IVA. The results of an interim analysis were consistent with previous RCTs, demonstrating both the safety and the sustained efficacy (24–36 weeks) of ELX/TEZ/IVA in pwCF 12 years or older homozygous for the F508del mutation or carrying an F508del and a minimal function mutation [69]. The effectiveness of triple combination therapy was also evaluated, in comparison to IVA or TEZ/IVA, in patients with F508del–gating or F508del–residual function mutations [23]. In both genotype groups, the triple combination regimen was associated with a marked reduction in sweat chloride levels and with a clear improvement in lung function (increased ppFEV_1_ and decreased pulmonary exacerbations), CFQ-R scores, and BMI (Table 2).

Considering the major improvements obtained with ELX/TEZ/IVA in pwCF aged 12 and older, the safety and efficacy of ELX/TEZ/IVA were recently evaluated in a multicentric phase 3 study including 6–11 year-old-children homozygous for the F508del or carrying an F508del-minimal function mutation [70]. By 24 weeks of treatment, children treated with ELX/TEZ/IVA had improved ppFEV1 and LCI2.5 in both genotype cohorts. Of note, sweat chloride concentrations decrease was greater in patients with F508del/F508del genotype than F508del/MF (70.4 mmol/L vs. 55.1 mmol/L).

The rate of AEs was comparable in the placebo and treatment groups, regardless of genotype or age [17,18,19,23,70], and included cough, increased sputum production, nasopharyngitis, upper respiratory tract infections, oropharyngeal pain, and fever, with no acute bronchoconstriction episodes reported. The most frequent laboratory abnormalities were elevated liver enzymes and bilirubin. ELX/TEZ/IVA discontinuation was limited and ranged between 1.5 and 9.5%.

### 4.2. Elexacaftor-Tezacaftor-Ivacaftor Combination in the French Real-World Studies

The elexacaftor–tezacaftor–ivacaftor combination was made available in France through an early access program for patients with advanced lung disease at the end of 2019. It has provided the opportunity to evaluate its safety and effectiveness in patients with advanced lung disease and who are at high risk of drug-related adverse effects and complications. In a study following 245 patients with a least one F508del mutation and advanced CF lung disease (median ppFEV_1_ 29, IQR 24–34), Burgel et al. found a mean improvement in ppFEV_1_ of +15.1 points after the initiation of ELX/TEZ/IVA; body weight increased by 4.2 kg on average [44]. The number of patients requiring long-term oxygen therapy, non-invasive ventilation, and/or enteral tube feeding decreased by 50%, 30%, and 50%, respectively. Only mild adverse effects, which did not require treatment discontinuation, were reported [44]. Furthermore, a two-fold decrease in the number of lung transplantations in pwCF between 2020 and the two previous years was found, suggesting that triple therapy has the potential to improve survival and delay the need for lung transplantation (LTx) [44].

These findings were confirmed by Martin et al. (2022) in a study evaluating LTx eligibility criteria and changes in lung function, nutritional status, healthcare resource utilization, and concurrent treatments over 12 months after the initiation of ELX/TEZ/IVA [46]. At baseline, 17 patients were waitlisted for transplantation, and 48 were considered for LTx within 3 months. At 1 month, ppFEV_1_ had increased by +13.4 percentage points (*p* < 0.0001) and remained stable throughout the 12-month observation period. After 1 year, 2 patients had been transplanted, 2 were still on the waiting list, and 61 no longer met transplantation criteria. Improvement in treatment burden decreased significantly, with an 86% decrease in the need for intravenous antibiotics, 59% for oxygen therapy, and 62% for non-invasive ventilation. ELX/TEZ/IVA treatment was well-tolerated, and no discontinuation was reported in this very fragile population. These findings are the first published evidence of prolonged disease modification in patients with advanced pulmonary disease who met the eligibility criteria for lung transplantation [46]. Interestingly, recent data from the French Agence de la Biomédecine, which monitors organ transplantation at the national level, showed a major decrease in lung transplantation for pwCF since the release of ELX/TEZ/IVA [71]. The French ELX/TEZ/IVA real-word studies are ongoing and include all children and adults with CF receiving this modulator combination ELX/TEZ/IVA nationally. Large-scale longer-term data are expected in future years and will hopefully confirm the promising clinical results that have been reported so far.

Recently, the results obtained with the French national cohort were also confirmed in the PROMISE study in which487 pwCF aged 12 years and older with at least 1 F508del allele and starting ETI for the first time were enrolled at 56 U.S. CF Foundation Therapeutics Development Network sites between November 2019 and May 2020 [72]. At 6 months and compared to baseline, authors reported a 9.76 percentage point increase in ppFEV_1_, +20.4 points in CFQR, a 41.7 mmol/L sweat chloride decrease, and a BMI increase. Of note, 44.1% entered the study using TEZ/IVA or LUM/IVA, whereas 6.7% were using IVA. Changes were larger in those naive to modulators but substantial in all groups, including those treated with IVA at baseline [72].

Martin et al. (2021) explored the patient’s perspectives in 101 patients with advanced CF aged 12 years and older who were treated with ELX/TEZ/IVA, using a 13-item online questionnaire that included 4 open questions [45]. Their goal was to gain a better understanding of how patients perceive the changes in respiratory symptoms, systemic manifestations, treatment burden, and overall quality of life. The authors reported that ELX/TEZ/IVA was associated with a significant improvement in respiratory symptoms, sleep quality, and physical self-esteem and with a reduction in treatment burden (chest physiotherapy, IV antibiotic courses, hospitalizations, diabetes control, concurrent treatments, lung transplant criteria). Furthermore, initiation of ELX/TEZ/IVA was associated with positive physical, psychological, and social effects, which translated into improved quality of life [45].

## 5. Conclusions and Perspectives

CF care has dramatically improved over the past 60 years. The introduction of highly effective CFTR modulators, to which a growing number of pwCF have become eligible over the past 10 years, is expected to have a profound impact on the prognosis and manifestations of CF. However, the survival gap between pwCF and the general population remains. Whether the development of highly effective CFTR modulators will reduce this gap remains to be seen. In the coming years, post-marketing real-world studies will play a key role in advancing our understanding of key pathophysiological changes in CF, identifying aspects of the disease that may be reversible, and providing important long-term data on the safety and effectiveness of CFTR modulators. Their use in specific populations, including patients with liver cirrhosis (which is present in 5–10% of pwCF) and in solid organ transplant recipients (lung, liver, and/or kidney), will need to be explored in real-world studies. Finally, the impact of CFTR modulators on emerging complications of CF such as colorectal cancer, cardiovascular disease, and chronic kidney disease remains unknown and should be examined in the coming years.

## Figures and Tables

**Figure 1 cells-11-01769-f001:**
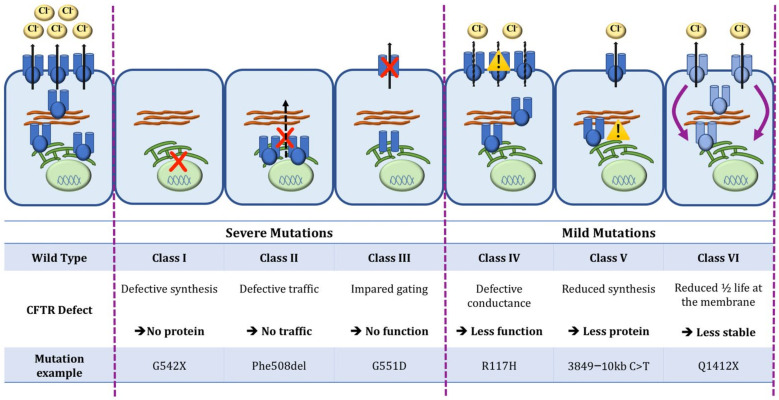
Classification of *CFTR* mutations. CFTR protein is located at the apical surface of epithelial cells, where it acts as a bicarbonate and chloride (Cl^−^) channel. Mutations in the *CFTR* gene are classified as severe (Classes I, II, and III), resulting in absent or minimal CFTR function, and mild (Classes IV, V, VI), usually with residual CFTR function.

**Figure 2 cells-11-01769-f002:**
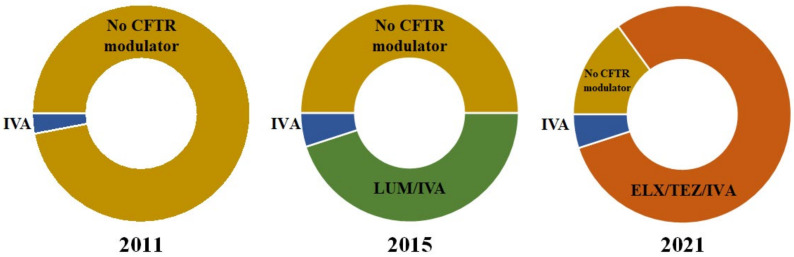
Proportion of the French CF population aged 12 years and older eligible for CFTR modulator therapy in 2011, 2015, and 2021 [20]. In 2011, only 3% of people with CF (pwCF) were eligible to receive a CFTR modulator (ivacaftor). In 2015, with lumacaftor–ivacaftor, half of the patient population became eligible, with 5% eligible for ivacaftor (at least one gating mutation) and 45% for lumacaftor–ivacaftor combination therapy (homozygous for the F508del mutation). By 2021, 82% were eligible for elexacaftor/tezacaftor/ivacaftor (at least one F508del mutation). There are still 10–15% of pwCF who have no access to CFTR modulator therapy. IVA: ivacaftor; LUM: lumacaftor; ELX: elexacaftor; TEZ: tezacaftor.

**Figure 3 cells-11-01769-f003:**
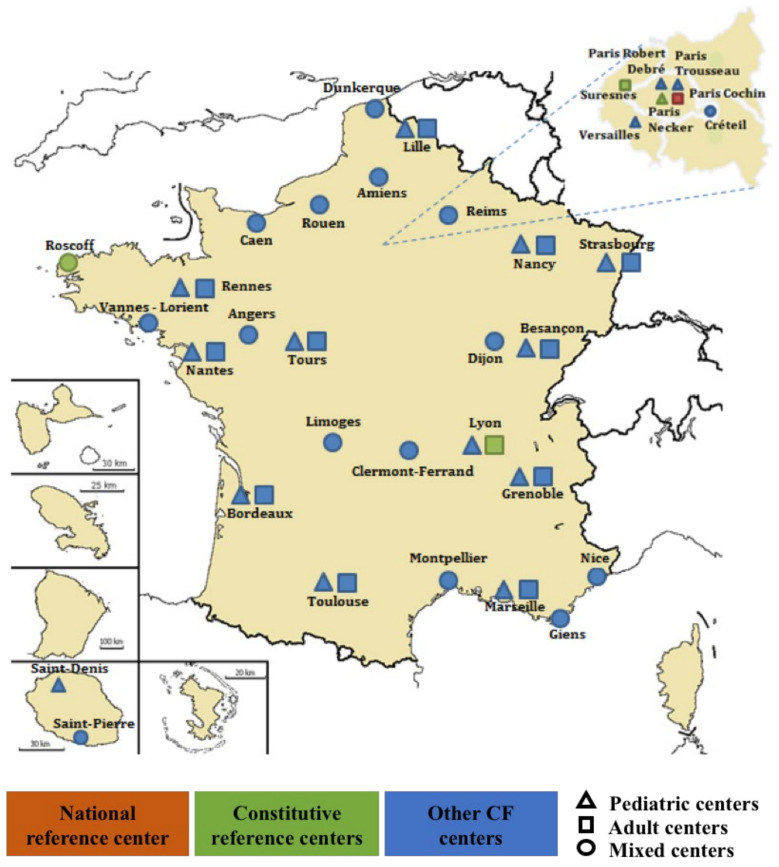
The French Cystic Fibrosis Reference Network.Upper insert: greater Paris area. Bottom left insert: La Réunion island.

**Table 1 cells-11-01769-t001:** Approved CFTR modulators and their indications in France (as of December 2021).

Modulator	Approval Year	Approved Ages	Target Mutations
**Ivacaftor**	2012	≥6 years	At least one copy of the G551D mutation
	2014	≥6 years	At least one gating (class III) mutation: G551D, G1244E, G1349D, G178R, G551S, S1251N, S1255P, S549N or S549R
	2016	≥2 years	
	2019	≥1 year	
	2020	≥6 months	
	2021	≥4 months	At least one gating (class III) mutation: G1244E, G1349D, G178R, G551D, S1251N, S1255P, S549N, S549R or G970R Or at least one copy of the R117H mutation
**Lumacaftor** **+** **Ivacaftor**	2016	≥12 years	Two copies of the F508del mutation
	2018	≥6 years	
	2019	≥2 years	
**Tezacaftor** **+** **Ivacaftor**	2020	≥12 years	Two copies of the F508del mutation Or One copy of the F508del mutation AND one of the following mutations: P67L, R117C, L206W, R352Q, A455E, D579G, 711+3A→G, S945L, S977F, R1070W, D1152H, 2789+5G→A, 3272 26A→G, 3849+10kbC→T.
	2021	≥6 years	
**Ivacaftor** **+** **Tezacaftor** **+** **Elexacaftor**	2020	≥12 years	Two copies of the F508del mutation Or One copy of the F508del mutation and one minimal function mutation
	2021	≥12 years	At least one F508del mutation

**Table 2 cells-11-01769-t002:** Summary of the main phase 3 randomized controlled trials of CFTR modulators in adolescents and adults with CF.

Study			Population				Outcomes				
Modulator	Author Year	Duration	*n*	Genotype	Age (Years) Mean [Range] *% <18 yrs*	ppFEV_1_ Range	Δ ppFEV_1_ (%)	Δ Sweat Cl^−^ (mmol/L)	Nutritional Changes	Δ CFQ-R Score (Points)	Discontinuation Rate
**Ivacaftor**	Ramsey 2011, [13]	48 weeks	167	≥1 G551D mutation	25.5 [12–53] *22.0%*	40–90	+10.6 *	−47.9 *	Weight +2.7 kg *	+ 8.6*	1%
	De Boeck 2014, [21]	8 weeks (Part 1)	39	≥1 non-G551D gating mutation	22.8 [6–57] *NA*	≥40	+10.7 *	−49.2 *	BMI +0.7 kg/m^2^ *	+ 9.6 *	7.7%
	Moss 2015, [22]	24 weeks	69	≥one *R117H* mutation	31.0 *[NA]* *27.5%*	≥40	+2.1 *	−24.0 *	BMI +0.26 kg/m^2^ *	+ 8.4 *	2.9%
**Ivacaftor** **+** **Lumacaftor**	Wainwright 2015, [14]	24 weeks	1108	Homozygous for F508del	25.1 [12–64] *26.1%*	40–90	+3.3 * & +2.8 *^#^	NA	BMI +0.28 * & + 0.24 kg/m^2^ *^#^	+ 3.1 * & +2.2 *^#^	4.2%
**Ivacaftor** **+** **Tezacaftor** **+** **Elexacaftor**	Heijerman 2019, [17]	4 weeks	107	Homozygous for F508del	28.4 *[NA]* *28.0%*	40–90	+10.0 ^±^	−45.1 ^±^	BMI +0.6 kg/m^2 ±^ Weight 1.6 kg ^±^	+ 17.4 ^±^	0%
	Middleton 2019, [18]	24 weeks	403	F508del-MF	26.2 *[NA]* *28.8%*	40–90	+14.3 *	−41.8 *	BMI +1.4 kg/m^2^ *	+ 20.2 *	1.5%
	Barry 2021, [23]	8 weeks	258	F508del -RF Or F508del-gating	37.7 *[NA]* *9.3%*	40–90	+3.5 ^∑^	−23.1 ^∑^	NA	+ 8.7 ^∑^	1.5%

Yrs: year; ppFEV_1_: percent predicted Forced Expiratory Volume in 1 s; Cl-: Chloride; CFQ-R: Cystic Fibrosis Questionnaire-Revised; BMI: body mass index; MF: minimal function; RF: residual function; NA: not available. ^#^: for the 600 mg daily of lumacaftor group and the 400 mg bid, respectively. *: compared to placebo; ^±^: compared to active control tezacaftor/ivacaftor ^∑^: compared to active control ivacaftor (F508del -RF) or tezacaftor/ivacaftor (F508del -gating mutation).

## Data Availability

Not applicable.

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
