# Peer review of "CFTR Modulators in People with Cystic Fibrosis: Real-World Evidence in France"

_cells, 2022, doi:10.3390/cells11111769_

Round 1

Reviewer 1 Report

This is a very interesting review describing the results of real-world studies on CFTR modulators conducted in the French CF network. I agree with the authors that real-world studies are very important to understand the benefit for patients outside the eligibility criteria for clinical trials. The description on the real-world studies conducted in France is quite complete, however, these results should be discussed in the context of other real-world studies. Further, the manuscript contains some typos and language quality could be improved throughout.

Major comments:

The authors have sorted the review by clinical trials on each modulator/modulator combination followed by real-world studies on each modulator. I suggest to re-order this review by CFTR modulator and describe the real world studies right after the clinical trial results to highlight the importance of real world studies and to underline the additional information gained by them.

For IVA, the authors present a discussion of their real-world studies in France compared to other published real-world studies. However, this discussion is completely missing for LUM/IVA and ELX/TEZ/IVA. I therefore suggest that the results of the real word studies on LUM/IVA and ELX/TEZ/IVA should be discussed in the context of other real-world studies in detail.

I suggest to also include the pivotal phase 3 trials in pediatric patients for the CFTR modulator/modulator combinations in the clinical trial description as some of the real world studies also include paediatric patients.

Minor comments:

Mutations names should be consistent throughout using either the 1 Letter code for aminoc acids (G551D) or the 3 letter abbreviation (Phe508del)

There are several typos throughout the manuscript (for example line 115, line 138)

Figure 1: I would suggest to remove the cilia from the cell model as the CFTR channel mainly presents it secretory cells or ionocytes rather than ciliated cells.

Line: 67: I suggest to remove the phrase “mutation-specific molecules” as current CFTR modulators are not mutation-specific.

Table 1: On what outcome measures is the interpretation of “moderate” and “high efficacy” based on? I suggest to remove the column “efficacy” as the efficacy is described in table 2.

For ELX/TEZ/IVA: 2020: “Two copies of the Phe508del mutation” is missing

2021: “One copy of the Phe508del mutation and one gating (class III) or one residual function mutation” should be changed to “at least one Phe508del mutation” as referred to in the main text

Line 84-85: The authors state that “since their introduction, CFTR modulators have dramatically changed clinical care, introducing a fundamental shift in perspective for pwCF and their caregivers (Figure 2).” However, Figure 2 does not support this statement and I suggest a reference here rather than a link to the Figure.

Lines 86-88: The selection criteria of clinical trials should be shortly described here.

Figure 2: Is this original data based on the French registry? Please provide a reference.

Line 155-157: Please provide references to these studies.

The authors state that “the incidence of AEs was generally similar in the LUM/IVA and placebo groups”. However, in the LUM/IVA Phase 3 clinical trial, the adverse events dyspnea and chest tightness were reported more frequently and the discontinuation rate was higher in the LUM/IVA groups compared to the placebo group. This should be mentioned in this context.

Although no real world studies on TEZ/IVA were performed in France, a short description of the clinical trials of TEZ/IVA should be included as TEZ/IVA is referred to several times in this review and it is the basis of the triple combination

Lines 236-238: The inclusion of results of TEZ/IVA in this paragraph seems odd and should be included in a separate paragraph (see comment above).

Table 2 and 3: I suggest to include a column with the age range in the study and to change the “%12-17 years” to “%<18 years” since there are several studies including patients 6 years and older as well.

Table 3: What criteria for the order of the studies was used? I suggest sorting them by retrospective or prospective and the year conducted. Further, the layout of the table should be reworked, as it seems like it was distorted over the some pages. In addition, I suggest to consistently use 1 decimal digit for ppFEV1 throughout the manuscript.

Lines 363-370. Why are these results describe together with the CFTR biomarker study rather than in the “safety and effectiveness” or in the “morphological changes” paragraph as the include only clinical and morphological outcomes.

Line 420-423: The authors claim that “LCI is not an ideal outcome measure … in adolescent and young adult pwCF with more advanced lung disease”. However, two independent studies (Shaw et al., JCF 2020; Graeber et. al., Annals ATS 2021) showed that LCI was sensitive to detect improvement by LUM/IVA also in patients with advanced lung disease, whereas FEV1 did not show any improvements.

Reviewer 2 Report

The rationale for this review paper stems from the need to document the impact of CFTR modulators on long term treatment efficacy and safety in the broader population- a population that includes those with advanced lung disease, liver disease, renal insufficiency or chronic bacterial infection. The data analyzed  was from the French CF Registry.

Critique:  The authors provided a lengthy background, describing the clinical trials results for participants tested with IVA, LUM+IVA, TEZA+IVA and ELX/TEZ/IVA. The background information was conveyed largely in the descriptive manner.  This reviewer would have found a graphical summary of the numerical metrics - more helpful.

The authors summarized the real-world clinical observations for approved modulators in a largely descriptive manner as well.  Although interesting, I had the impression that this article was somewhat premature because the time for collecting such real-world data is still relatively short.  Meaningful mathematical modeling of changes to patient survival cannot yet be performed for ETI. Similarly, the impact of modulators on the prevention of CF associated diseases, such as liver cirrhosis, colorectal cancer, cardiovascular and chronic renal diseases cannot yet be evaluated in a quantitative manner.

In summary, I had the impression that more time is needed to curate clinical responses to modulators, thereby enabling greater insight into long term treatment effectiveness.

Reviewer 3 Report

Regard and collegues present a review manuscript wiht a clearly defined focus on CFTR modulator studies conducted in France. While, as normal for a structured review, all findings presented have been published elsewhere before, Regard and colleagues provide a very organized overview which would be rather cumbersome for any interested party to compile.

The reviewer has no suggestions to improve this well-written manuscript and thus, recommends to accept the manuscript in its present form.

For the sake of transparency, the reviewer assumes that all text is devoid of plagiarisms, but the reviewer did not check for duplicate phrasing with google as many KI systems are better to detect duplicated text and the reviewer asumes that MDPI will employ those.

Round 2

Reviewer 1 Report

The authors have responded adequately to all the reviewers comments and have made significant adjustments to the manuscript.